# Phenotypic plasticity as a mechanism of cave colonization and adaptation

Helena Bilandžija[1,2], Breanna Hollifield[1], Mireille Steck[3], Guanliang Meng[4,5], Mandy Ng[1], Andrew D Koch[6], Romana Gračan[7], Helena Ćetković[2], Megan L Porter[3], Kenneth J Renner[6], William Jeffery[1]*

[1]Department of Biology, University of Maryland, College Park, United States; [2]Department of Molecular Biology, Ruđer Bošković Institute, Zagreb, Croatia; [3]Department of Biology, University of Hawai'i at Mānoa, Honolulu, United States; [4]BGI-Shenzhen, Shenzhen, China; [5]China National GeneBank, BGI-Shenzhen, Shenzhen, China; [6]Department of Biology, University of South Dakota, Vermillion, United States; [7]Department of Biology, Faculty of Science, University of Zagreb, Zagreb, Croatia

**Abstract** A widely accepted model for the evolution of cave animals posits colonization by surface ancestors followed by the acquisition of adaptations over many generations. However, the speed of cave adaptation in some species suggests mechanisms operating over shorter timescales. To address these mechanisms, we used *Astyanax mexicanus*, a teleost with ancestral surface morphs (surface fish, SF) and derived cave morphs (cavefish, CF). We exposed SF to completely dark conditions and identified numerous altered traits at both the gene expression and phenotypic levels. Remarkably, most of these alterations mimicked CF phenotypes. Our results indicate that many cave-related traits can appear within a single generation by phenotypic plasticity. In the next generation, plasticity can be further refined. The initial plastic responses are random in adaptive outcome but may determine the subsequent course of evolution. Our study suggests that phenotypic plasticity contributes to the rapid evolution of cave-related traits in *A. mexicanus*.

**\*For correspondence:**
Jeffery@umd.edu

**Competing interests:** The authors declare that no competing interests exist.

## Introduction

A major problem in modern biology is understanding how organisms adapt to an environmental change and how complex, adaptive phenotypes originate. Phenotypic evolution can result from standing genetic variation, new mutations, or phenotypic plasticity followed by genetic assimilation, but these processes are often difficult to distinguish in slowly changing environments. This difficulty can be overcome by studying adaptation to more abrupt environmental changes, such as the dramatic transition from life on the Earth's surface to subterranean voids and caves.

A unifying feature of subterranean environments is complete darkness (*Culver and Pipan, 2009*; *Pipan and Culver, 2012*). Cave-adapted animals have evolved a range of unusual and specialized traits, often called troglomorphic traits, which enable survival in challenging conditions of the subsurface. In cave dwelling animals, visual senses and protection from the effects of sunlight are unnecessary, and consequently eyes and pigmentation are usually reduced or absent. To compensate for lack of vision, other traits, especially those related to chemo- and mechano-receptor sensations, are enhanced. Circadian rhythms that fine-tune organismal physiology with day-night cycles are also distorted, and light dependent behaviors, as well as the neural and endocrine circuits controlling these behaviors, are modified. Because photosynthetic organisms are not present in caves, primary productivity is absent and nutrient availability is usually limited. Survival under conditions of reduced and/or sporadic food resources is possible due to the evolution of modified feeding behaviors and

**eLife digest** The Mexican tetra is a fish that has two forms: a surface-dwelling form, which has eyes and silvery grey appearance, and a cave-dwelling form, which is blind and has lost its pigmentation. Recent studies have shown that the cave-dwelling form evolved rapidly within the last 200,000 years from an ancestor that lived at the surface. The recent evolution of the cave-dwelling form of the tetra poses an interesting evolutionary question: how did the surface-dwelling ancestor of the tetra quickly adapt to the new and challenging environment found in the caves?

'Phenotypic plasticity' is a phenomenon through which a single set of genes can produce different observable traits depending on the environment. An example of phenotypic plasticity occurs in response to diet: in animals, poor diets can lead to an increase in the size of the digestive organs and to the animals eating more. To see if surface-dwelling tetras can quickly adapt to cave environments through phenotypic plasticity, Bilandžija et al. have exposed these fish to complete darkness (the major feature of the cave environment) for two years. After spending up to two years in the dark, these fish were compared to normal surface-dwelling and cave-dwelling tetras.

Results revealed that surface-dwelling tetras raised in the dark exhibited traits associated with cave-dwelling tetras. These traits included changes in the activity of many genes involved in diverse processes, resistance to starvation, metabolism, and levels of hormones and molecules involved in neural signaling, which could lead to changes in behavior. However, the fish also exhibited traits, including an increase in the cells responsible for pigmentation, that would have no obvious benefit in the darkness. Even though the changes observed require no genetic mutations, they can help or hinder the fish's survival once they occur, possibly determining subsequent evolution. Thus, a trait beneficial for surviving in the dark that appears simply through phenotypic plasticity may eventually be selected for and genetic mutations that encode it more reliably may appear too.

These results shed light on how species may quickly adapt to new environments without accumulating genetic mutations, which can take hundreds of thousands of years. They also may help to explain how colonizer species succeed in challenging environments. The principles described by Bilandžija et al. can be applied to different organisms adapting to new environments, and may help understand the role of phenotypic plasticity in evolution.

adaptive changes in metabolism, such as lower metabolic rate, increased starvation resistance, and changes in carbohydrate and lipid metabolism (*Culver and Pipan, 2009*).

The ancestors of cave dwelling animals originally lived on the surface. Regardless of whether the pioneering animals entered the subsurface accidently (by capture of surface waters or by falling into a pit) or purposely (to take refuge from harsh competition or changing climate conditions), they were suddenly exposed to darkness and subterranean living conditions. How did they overcome the challenges of life in darkness, such as orientation in the absence of light, survival during long periods without food, and detection of mates, in order to successfully establish underground lineages? The preadaptation hypothesis suggests that the ancestors of cave adapted animals were pre-adapted to darkness by already leading semi-nocturnal lifestyles in environments with low-light penetration, such as rotting logs, leaf litter, or in the soil, simplifying their transition to life in caves (*Vandel, 1965*). However, many cave dwelling animals, such as the cavefish *Astyanax mexicanus*, are descended from diurnal surface-dwelling animals; and the cavefish *Poecillia mexicana*, is derived from photophilic surface ancestors (*Parzefall et al., 2007*) with no obvious pre-adaptations to low light. For these animals, the transition to dark subterranean habitats could have posed significant challenges.

In the present investigation, we used the teleost *Astyanax mexicanus* as a model system to understand the molecular, physiological and morphological basis of cave colonization. *A. mexicanus* is an excellent system for studying adaptation to cave life because it consists of both the ancestral-proxy form and multiple derived forms: surface dwelling (surface fish, SF) and cave-dwelling conspecifics (cavefish, CF) respectively. Surface fish live in rivers exposed to day-night cycles with abundant food availability. In the karstic Sierra Madre Oriental del Norte mountain range in Mexico, SF have colonized about 30 caves (*Gross, 2012*), and their cave dwelling descendants have adapted to permanently dark, mostly food restricted subterranean waters.

Recent studies revealed that cavefish lineages are much younger than previously thought, from less than 20,000 (*Fumey et al., 2018*) to about 200,000 (*Herman et al., 2018*) years old, implying that the evolution of numerous cave-related phenotypes may have occurred over relatively short time periods. Furthermore, troglomorphic phenotypes are maintained in *A. mexicanus* cavefish despite the lack of complete isolation between surface fish and cavefish populations (*Bradic et al., 2012*; *Chakraborty and Nei, 1974*; *Herman et al., 2018*; *Wilkens and Hüppop, 1986*). A remarkably recent origin of troglomorphic traits, also involving likely immigration from surface populations, has also been proposed for the evolution of cave-related traits in other cave dwelling animals, including invertebrates and vertebrates (*Behrmann-Godel et al., 2017*; *Klaus et al., 2013*; *Niemiller et al., 2008*; *Schilthuizen et al., 2005*; *Villacorta et al., 2008*; *Zhang and Li, 2013*). What are the mechanisms that enable the rapid evolution of various distinct cave-related phenotypes, and how are they preserved despite the homogenizing process of gene flow from surface populations? Clearly, strong selection must be involved in these cases, but in order for selection to act, it is essential to have phenotypic variants with differences in fitness in the cave environment.

The conspecific surface and cave morphs of *A. mexicanus* are capable of hybridization in the laboratory and thus have been the subjects of extensive genetic analysis (*O'Quin et al., 2013*; *Protas et al., 2008*; *Yoshizawa et al., 2012*). However, not many mutations have been uncovered in CF, even in the presumably dispensable genes associated with eye degeneration, such as crystallin genes (*Hinaux et al., 2013*; *Ma et al., 2014*). A more immediate mechanism for survival is likely necessary when new colonizers first enter caves. This suggests the presence of standing genetic variation and/or phenotypic plasticity followed by genetic assimilation (*Waddington, 1953*), rather than the complete reliance on the accumulation of new mutations to produce the necessary variability. An instance of standing genetic variation has been reported in *A. mexicanus* CF with respect to the evolution of eye reduction (*Rohner et al., 2013*), but phenotypic plasticity has not been studied extensively in this system (but see *Reyes, 2015*). This is surprising since phenotypic plasticity plays an important role in the adaptation of many different organisms to changing environments and the colonization of new habitats reviewed in *Fox et al. (2019)*; *Morris (2014)*.

To replicate the colonization of the subterranean environment, we placed *A. mexicanus* SF in completely dark conditions for up to two years.

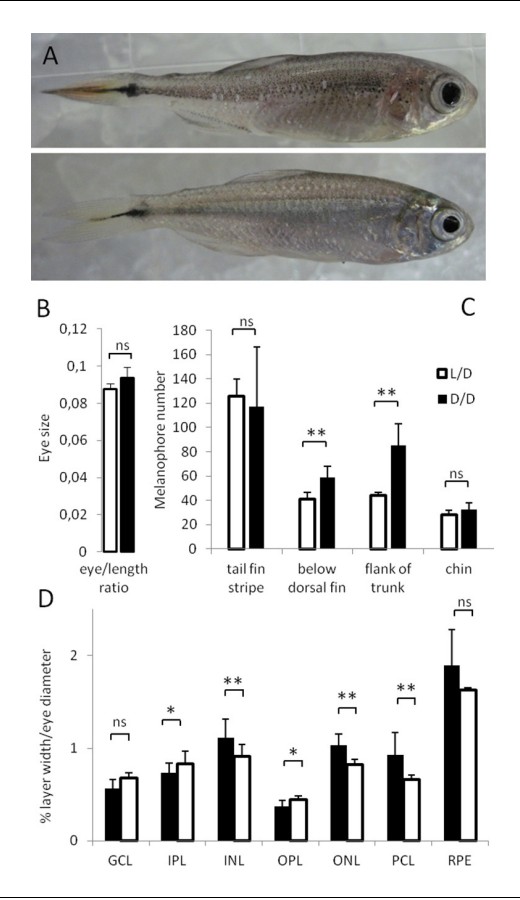

**Figure 1.** Morphological differences in *Astyanax mexicanus* surface fish maintained in different light regimes. (**A**) Surface fish (SF) kept in constant dark (D/D; top frame) *vs.* light/dark (L/D; bottom frame) photoperiod for 1 year. (**B**) Eye size normalized by body length in D/D vs. L/D SF kept in the experimental conditions for 1 to 2 years. (N = 8) (**C**) Number of melanophores in 1 year-old D/D vs. L/D SF determined in four different body regions. (N = 5) (**D**) Thickness of retinal layers in D/D (N = 4) vs. L/D fish (N = 3): GCL, ganglion cell layer; IPL, inner plexiform layer; INL, inner nuclear layer; OPL, outer plexiform layer; ONL, outer nuclear layer; PCL photoreceptor cell layer; RPE, retinal pigment epithelium measured as a ratio to eye diameter. (Error bars: SD; T-test Ns – not significant, *p<0.05, **p<0.001). *Figure 1—source data 1* contains raw data and summary statistics.

The online version of this article includes the following source data and figure supplement(s) for figure 1:

**Source data 1.** Raw data and summary statistics for *Figure 1*.

**Figure supplement 1.** Cross section of retinal layers in surface fish reared in L/D (top) or D/D (bottom) conditions.

By assaying different traits, we discovered that dark-raised SF show numerous phenotypes that are normally associated with CF adaptations to dark environments. The results imply that some cave-adapted traits may have appeared rapidly by phenotypic plasticity in *A. mexicanus*. Our results provide a basis for the evolution of adaptations in the cavefish lineage by genetic assimilation.

## Results

### Morphological changes in dark-raised surface fish

In this study, we exposed SF to D/D from early developmental stages until up to 2 years after spawning, and compared them to L/D controls. As a first step we compared body shape, length and width, eye size, and pigmentation in SF raised under D/D and L/D conditions (*Figure 1A*). We found no consistent differences in body parameters or eye sizes between D/D and L/D SF (*Figure 1B*). However, when we compared thickness of retinal layers between the two groups, 5 out of 7 layers were significantly different: D/D fish showed a thinning of the two plexiform layers and a thickening of the two nuclear layers and the photoreceptor layer (*Figure 1D*, *Figure 1—figure supplement 1*). Surprisingly, we also observed a significantly higher number of melanophores in the flank of the trunk and below the dorsal fin in D/D compared to L/D SF, although similar levels of pigmentation were noted in other regions of the body (*Figure 1C*).

### Gene expression changes in dark-raised surface fish

To understand the effects of exposure to constant darkness at the molecular level, a comparative transcriptomic analysis was conducted. We used three, 7 month-old surface fish placed in either D/D or L/D conditions within one dpf (day post fertilization). RNA sequencing yielded 21.9 to 27.6 million reads from each fish, representing a total of 25,194 genes. Basic statistical data are included in *Supplementary file 2*. We found 356 differentially expressed genes at a significance threshold of $p_{adj} < 0.1$ (*Figure 2—source data 1*). Of these, 210 were up-regulated and 146 were down-regulated in the D/D fish; the set of differentially expressed genes contained 67 genes with unconfirmed functions. We were particularly interested in the genes related to known aspects of the CF phenotype. Genes involved in circadian regulation, locomotor rhythm and visual perception were down-regulated in D/D fish, whereas lipid metabolism was the main functional category enriched in the up-regulated gene set. Two genes involved in pigmentation were also changed but their expression was higher in D/D, consistent with the increase in pigment cells described above. We also found significant changes in gene expression which were not predicted from known CF-associated phenotypes. These genes function in oxidation-reduction processes, hormone activity, hemostasis, aromatic amino acid metabolism, gene expression, metabolism of proteins, and signal transduction (*Figure 2*).

To validate the RNAseq results, we examined the levels of several differentially expressed genes by real time PCR (RT-PCR) (*Figure 3*). We confirmed the down-regulation of the hormone related genes *somatostatin 1 tandem duplicate 2* (*sst1.2*) and *inhibin beta B* (*inhbb*), ros involved *dual oxidase* (*duox*), the circadian rhythm genes *period circadian clock 2* (*per2*) and *cryptochrome-1-like* (*cry3b*), the vision related genes *retinoschisin 1a* (*rs1a*) and *tubby like protein 1a* (*tulp1a*), as well as genes involved in the regulation of metabolism *pancreatic and duodenal homeobox 1* (*pdx1*), and *deptor* in D/D compared to L/D SF. However, some genes, most notably aromatic amino acid metabolism genes *hydroxyphenylpyruvate dioxygenase a* (*hpda*), *tryptophan 2,3-dioxygenase* (*tdo2a*), *aralkylamine N-acetyltransferase 1* (*aanat1*), and *tryptophan hydroxylase 1a* (*tph1a*) showed down-regulation according to RT-PCR results, whereas they were up-regulated in the transcriptome. This discrepancy may be related to differences in the age or condition of fish at the time of sampling.

In addition, we quantified some genes which did not show significant differential expression according to RNAseq results but are important for the cavefish phenotype: *heat shock protein 90α* (*hsp90aa1.2*), and the de novo DNA methyltransferases *dnmt1*, *dnmt3aa*, *dnmt3ab*, *dnmt3ba* and *dnmt3bb*. *Hsp90aa1.2*, *dnmt1* and *dnmt3bb* showed changes in expression related to photoperiod in *Astyanax*.

The RNAseq results and subsequent RT-PCR validation showed changes in the expression of differentially expressed genes relative to photoperiod in both SF and PA CF. Expression of some genes

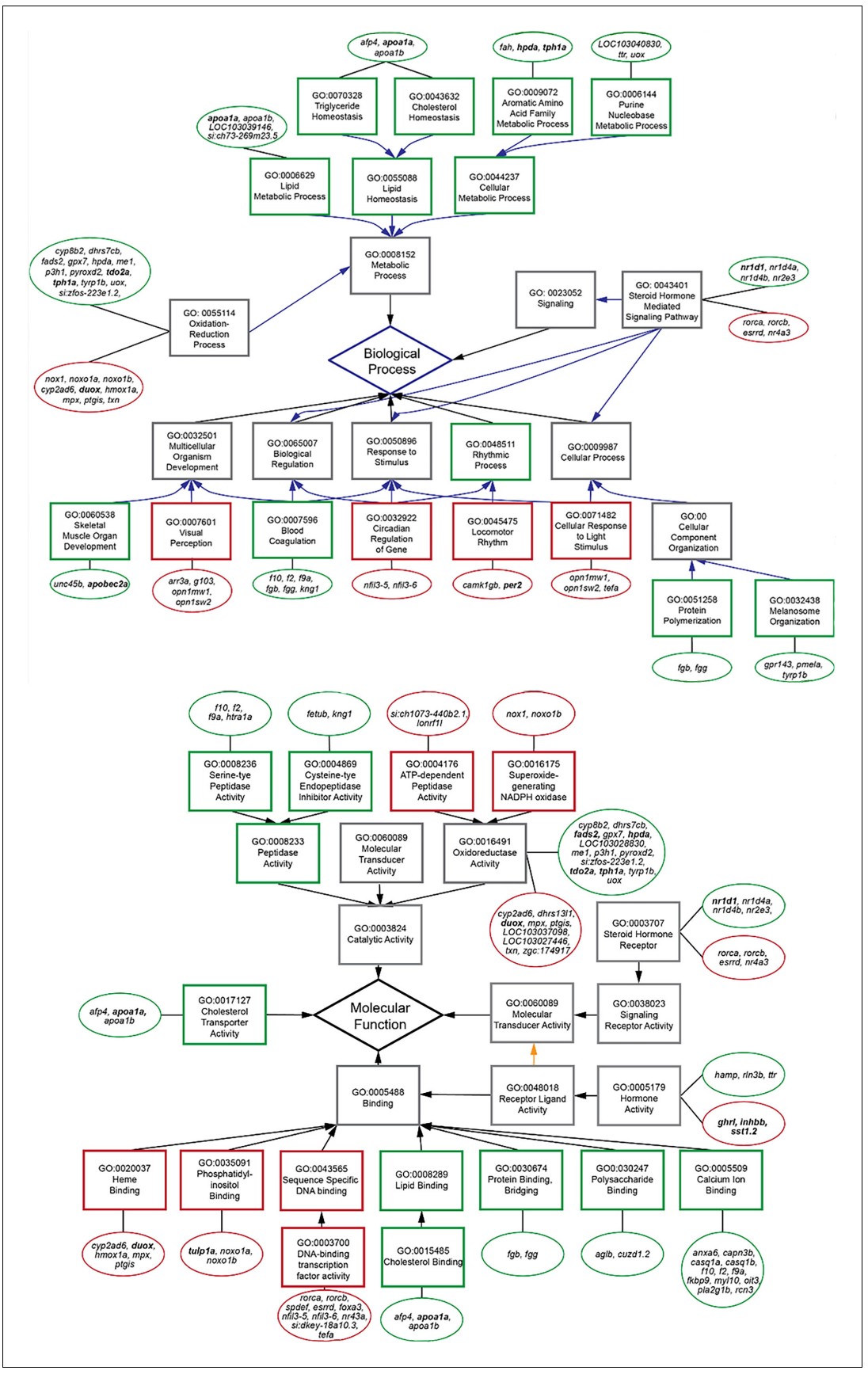

**Figure 2.** Subset of relevant, enriched GO terms (boxes) for biological processes (top) and molecular functions (bottom) of differentially expressed genes (circles) in the transcriptome. Genes tested by RT-PCR are in bold. Red outlines are down-regulated terms and genes, and green outlines are up-regulated.

The online version of this article includes the following source data for figure 2:

**Source data 1.** List of differentially expressed genes between surface fish maintained in different light regimes.

changed in the same direction in both SF and CF (e.g. *inhbb*, *per2*, *sst1.2*), whereas others changed in one and not the other fish type, or changed in the opposite directions in different fish types (e.g. *duox*, *nr1d1*, *tph1a*). Except for *duox* and *dnmt3bb.1*, all of the significantly changed genes show the same direction of changes in L/D vs. D/D SF and in SF vs. PA: *hsp90aa1.2*, *rs1a*, *tdo2a*, *dnmt1*, *per2*, *sst1.2*, *tulp1a*, *aanat1*, and *tph1a*.

## Starvation resistance in dark-raised surface fish

A major challenge facing animals that colonize caves is low food availability due to limited or absent primary productivity. To determine how a surface ancestor may have coped with this difficulty, we raised SF and PA larvae in D/D and L/D beginning < 24 hpf (hours post fertilization) without feeding (N = 36 larvae/group). D/D and L/D controls were fed daily portions of brine shrimp from seven dpf, when larvae normally lose their reliance on yolk and begin feeding. By 15 dpf, about 25% more SF and over 65% more PA unfed larvae were alive in D/D compared to L/D conditions. By 18 dpf, only a few unfed embryos were still alive, and they were all from the D/D conditions (8 PA and 2 SF) (**Figure 4**). In control conditions, SF survived better than PA, and both fish types had higher survival in D/D conditions. By 18dpf, 35 SF D/D, 17 SF L/D, 20 PA D/D and 12 PA L/D larvae were still alive. The results show that dark raised SF and CF are more resistant to starvation than siblings raised under a normal photoperiod and that PA larvae survive starvation better than SF larvae regardless of the lighting conditions.

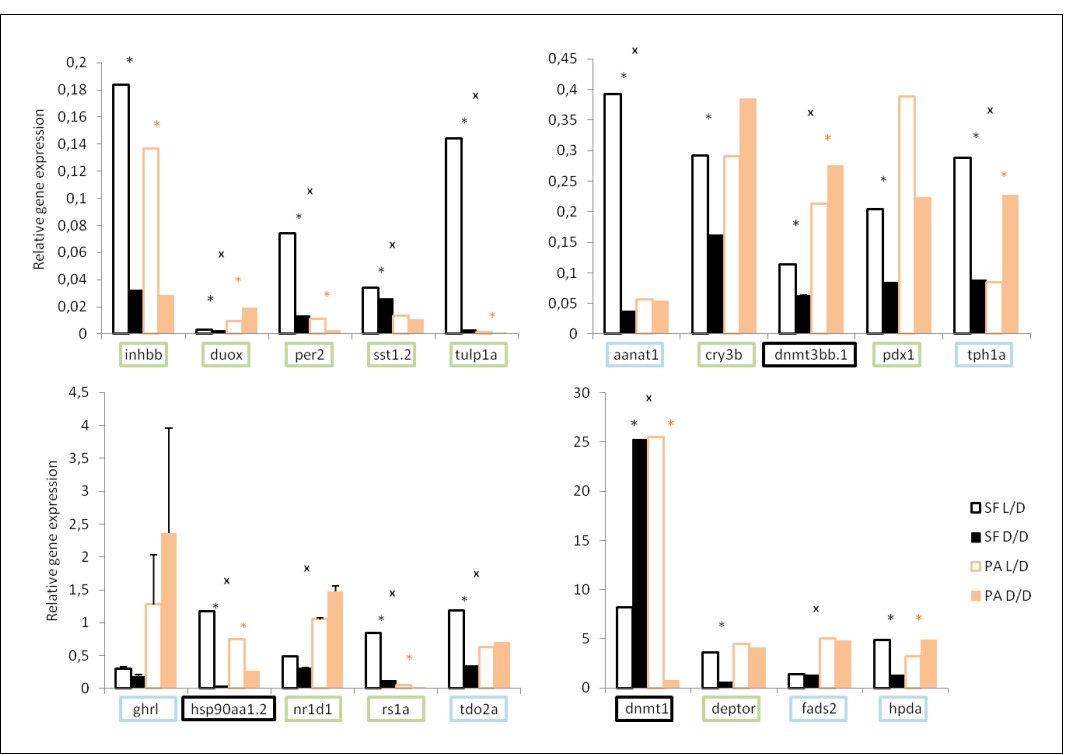

**Figure 3.** Normalized relative expression levels of genes in D/D and L/D SF and PA determined by RT-PCR. (Error bars: SD; ANOVA with Bonferroni adjustments p<0.05 black * for SF D/D vs. SF L/D; pink * for PA D/D vs. PA L/D; X for SF vs. PA,). Genes that showed the same direction of change in transcriptome and rtPCR are in green, genes that do not show the latter changes in blue, and genes chosen based on previous work in *Astyanax* in black.

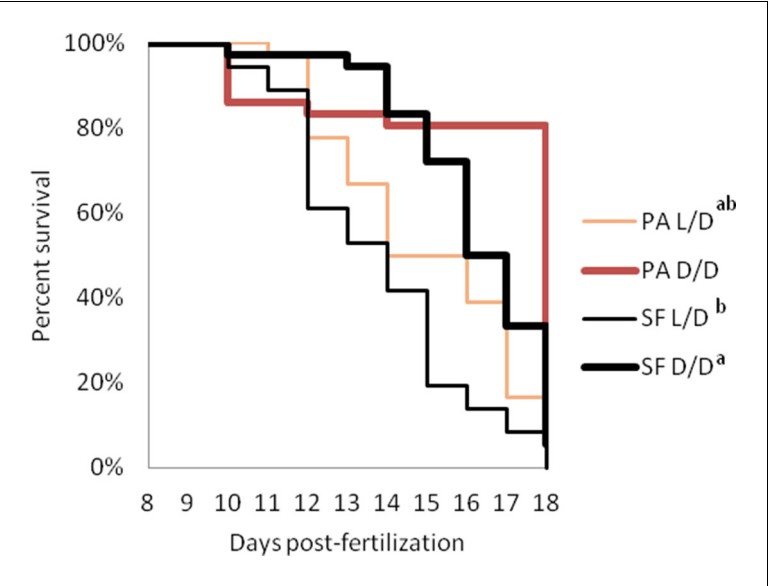

**Figure 4.** Survival curves of starvation resistance in *Astyanax mexicanus* surface fish (SF) and Pachón cavefish (PA) raised in complete darkness (D/D) or a normal photoperiod (L/D). Graphs show the percent of surviving fish (from the initial 36) on each day. Groups of SF and PA larvae from each condition were starved starting at seven dpf. Vertical drops represent individuals lost at a given time point. Groups in the legend that share a superscript are not statistically different, p values calculated by Cox proportional hazards model followed by generalized linear hypothesis test. *Figure 4—source data 1* contains raw data.
The online version of this article includes the following source data for figure 4:

**Source data 1.** Starvation survival of *Astyanax mexicanus* surface fish and Pachón cavefish raised in different light conditions.

## Metabolic rate decrease in dark-raised surface fish

We hypothesized that one of the reasons why fish in D/D survived starvation longer was because of lower energy expenditure. To test this possibility, we exposed SF and PA to D/D vs. L/D conditions within first 24 hpf after spawning and measured $O_2$ consumption at 2.5 dpf and 7.5 dpf. The results showed that SF and PA larvae raised in D/D have decreased metabolic rates when compared to SF and PA larvae raised in L/D conditions. This result was found to be significant in SF each time this experiment was conducted (three replicates) for both 2.5 and 7.5-day-old larvae. Furthermore, D/D PA showed a decrease in $O_2$ consumption at 2.5 but not at 7.5 dpf when compared to larvae raised in L/D. In addition, PA and SF metabolic rates were not significantly different at 2.5 dpf, whereas at 7.5 dpf PA has a lower metabolic rate than SF larvae raised in L/D conditions. At 2.5 dpf metabolic rate is similar between SF and PA, and darkness caused it to be reduced in both types of fish. However, by 7.5 dpf the metabolic rate in PA was reduced compared to L/D SF, and PA did not show a plastic response to darkness, whereas at the same age SF metabolic rate was still affected by darkness (*Figure 5*). In summary, the results support the hypothesis that D/D fish survive starvation longer because of lower metabolic rates. Improved survival in darkness also could be mediated by other factors.

## Cortisol increases in dark-raised surface fish

We hypothesized that exposure to constant darkness may represent a chronic stressor and tested this possibility by comparing cortisol levels in both L/D and D/D SF and PA. Surface fish raised in D/D conditions had significantly higher cortisol levels than L/D SF (*Figure 6*). In contrast, PA did not show a significant change in cortisol levels after exposure to D/D. This result was confirmed in three independent experiments using fish of different ages and different periods of dark exposure and shows that dark raised SF exhibit higher cortisol levels than SF raised on a normal photoperiod.

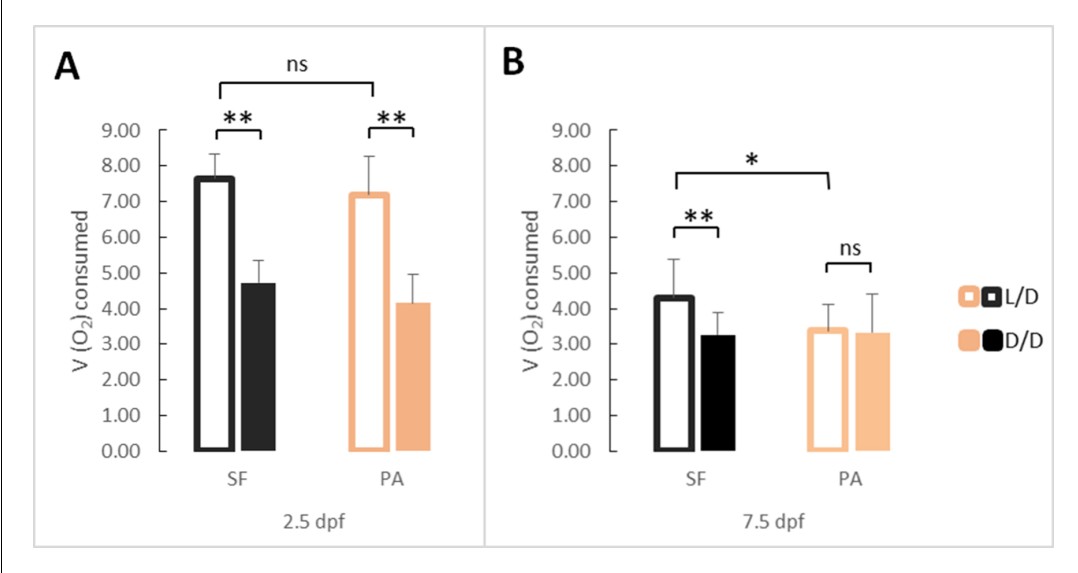

**Figure 5.** Average oxygen consumption at 2.5 (**A**) and 7.5 dpf (**B**) in SF and PA larvae kept in D/D versus L/D conditions. At 2.5 dpf N = 18 (SF L/D), 19 (SF D/D), 18 (PA L/D), 25 (PA D/D), and at 7.5 dpf N = 20 (SF L/D), 24 (SF D/D), 20 (PA L/D) and 24 (PA D/D). (Error bars represent standard deviation, ns: not significant, *p<0.05, **p<0.01 as calculated by ANOVA and Tukey HSD Test.) *Figure 5—source data 1* contains raw data and summary statistics.

The online version of this article includes the following source data for figure 5:

**Source data 1.** Raw data and statistics for *Figure 5*.

## Fat increase in dark-raised surface fish

Our RNAseq results show that genes involved in many aspects of fat metabolism were up-regulated in D/D versus L/D reared SF. To confirm this on a phenotypic level, we quantified triglyceride content in SF and PA raised to adulthood under D/D and L/D conditions. In fish raised under L/D conditions, PA had higher triglyceride levels than SF. Surface fish raised in D/D conditions had markedly higher triglyceride levels than L/D SF. Triglyceride levels in D/D PA were also higher than levels measured in L/D PA, although the difference is more modest than that observed for SF (*Figure 7*). These results suggest that dark raised SF have higher levels of triglyceride metabolism that SF raised on a normal photoperiod.

## Hormone levels change in dark-raised surface fish

Because of morphological changes in the pituitary and thyroid glands in dark raised SF (*Rasquin, 1949*) and differential expression of some of the genes associated with pituitary hormones (e.g. *sst1.2*, *inhbb*, *ghrl*) in the dark-raised SF transcriptome, we quantified the levels of the pituitary hormones thyroid stimulating hormone and growth hormone in SF and PA adults raised in either L/D or D/D conditions. Thyroid stimulating hormone levels were lower in D/D SF when compared to L/D SF and higher in D/D PA when compared to L/D PA, although the difference was not statistically different in each experimental replicate. PA had higher thyroid stimulating hormone levels than SF regardless of lighting conditions (*Figure 8A*). When thyroid stimulating hormone levels were compared between SF and three populations of CF, there was a trend for higher thyroid stimulating hormone levels in TI (p=0.07), but not MO (p=0.16), and thyroid stimulating hormone levels were significantly higher in PA (*Figure 8B*). The levels of growth hormone were higher in both SF and PA raised in D/D conditions when compared to fish raised in L/D (*Figure 9A*). All three populations of CF had higher growth hormone levels than SF (*Figure 9B*).

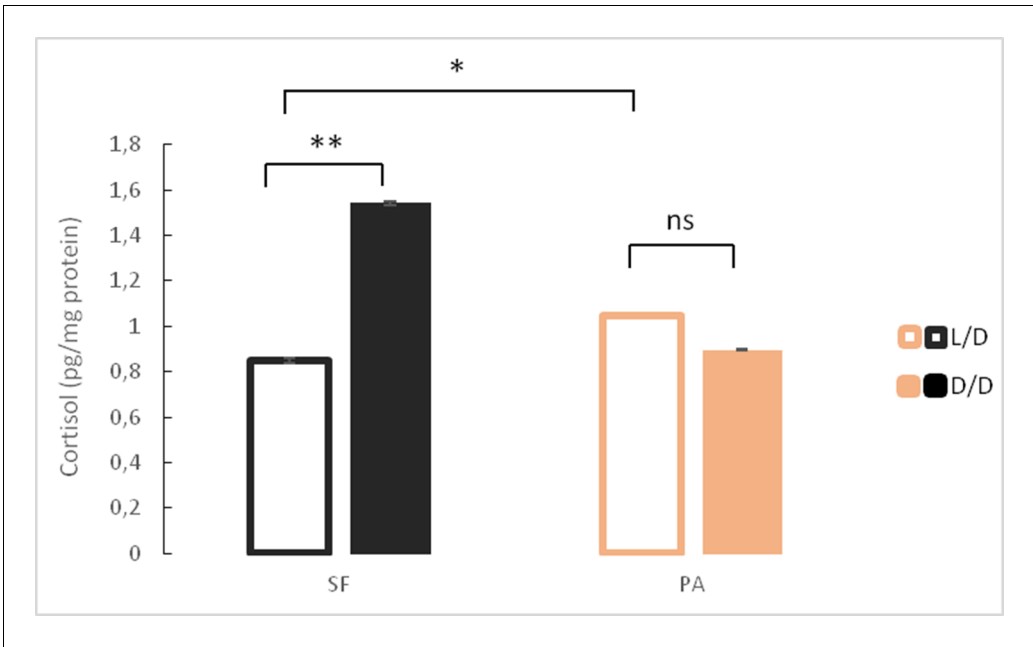

**Figure 6.** Mean cortisol levels in adult surface fish (SF) and Pachón cavefish (PA) kept in D/D or L/D conditions for 1.5 to 2 years (N = 4/group). (Error bars represent SD in three technical replicates, ANOVA and Tukey HSD Test: Ns – not significant, *p<0.05, **p<0.01). *Figure 6—source data 1* contains raw data and summary statistics. The online version of this article includes the following source data for figure 6:

**Source data 1.** Raw data and summary statistics for *Figure 6*.

## Serotonin decrease in dark-raised surface fish

According to the RNAseq analysis, the tryptophan metabolism pathway was significantly up-regulated in D/D versus L/D SF, while real time qPCR suggested lower expression of the *tdo2a*, *aanat1* and *tph1a* genes in dark–raised SF. Therefore, we compared serotonin (5-HT) levels in D/D and L/D SF and PA adults by HPLC. Since serotonin levels fluctuate according to a daily rhythm (*Fingerman, 1976*), samples for this assay were collected between 11 AM and 3 PM (day) or 11 PM and 3 AM (night) regardless of rearing conditions. Serotonin concentrations were lower in the brains (*Figure 10A*) and the bodies (*Figure 10B*) of D/D compared to L/D SF. Likewise, the 5-HT metabolite 5-HIAA was lower in the brain of D/D SF (*Figure 10—figure supplement 1*). It was not possible to quantify 5-HIAA in the body because it was masked by several interfering peaks. Serotonin levels in the SF brain, but not the body, were significantly lower at night compared to samples collected during the day. In PA brains collected during the day, 5-HT levels were lower in fish raised in D/D compared to L/D conditions but 5-HT levels were similar in D/D and L/D brains at night. Modest, but significant, changes in body 5-HT were evident in PA with increases in D/D relative to L/D collected during the day, and the opposite change occurred in bodies collected during the night. Serotonin levels were significantly lower in PA brains and bodies when compared to SF regardless of experimental conditions (D/D, L/D, day/night). Comparison of SF brain 5-HT with TI, MO and PA fish showed that all three cave populations have lower brain 5-HT (*Figure 10C*). In contrast to adults, 5-HT was significantly higher in PA compared to SF at seven dpf, and darkness did not affect 5-HT levels in either larval fish types (*Figure 10D*).

## Progeny of dark-raised surface fish show plasticity in starvation resistance and metabolic rate

To test whether the plastic changes in the fish reared in the dark during the first generation are maintained and transferred to the next generation, we induced spawning in D/D reared SF. We

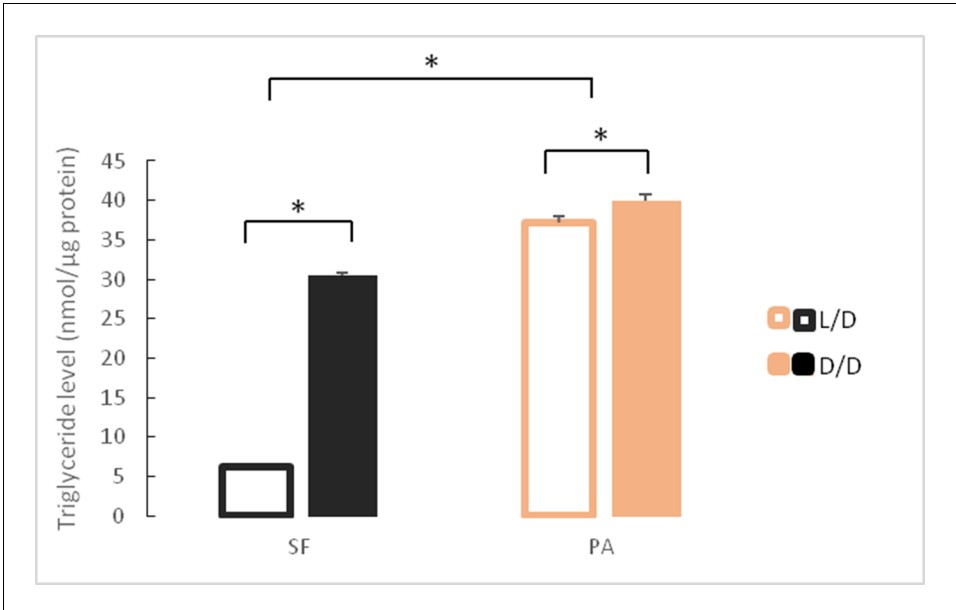

**Figure 7.** Mean triglyceride levels in SF and PA raised under D/D versus L/D conditions for approximately 1 year since < 24 hpf.  N = 3 fish/group (Error bars represent standard deviation. *p<0.01; ANOVA and Tukey HSD Test). *Figure 7—source data 1* contains raw data and summary statistics.

The online version of this article includes the following source data for figure 7:

**Source data 1.** Raw data and summary statistics for *Figure 7*.

measured starvation resistance and metabolic rate in G1 embryos (dSF) developing in D/D or in L/D during the second generation, as well as the offspring of L/D reared controls exposed to L/D or D/D conditions. The G1 larvae reared in L/D showed no differences in starvation resistance (*Figure 11*) or metabolic rate (*Figure 12*), regardless of whether they were the progeny of L/D or D/D parents. Their cohorts raised in darkness showed a modification in these phenotypes: dSF larvae derived

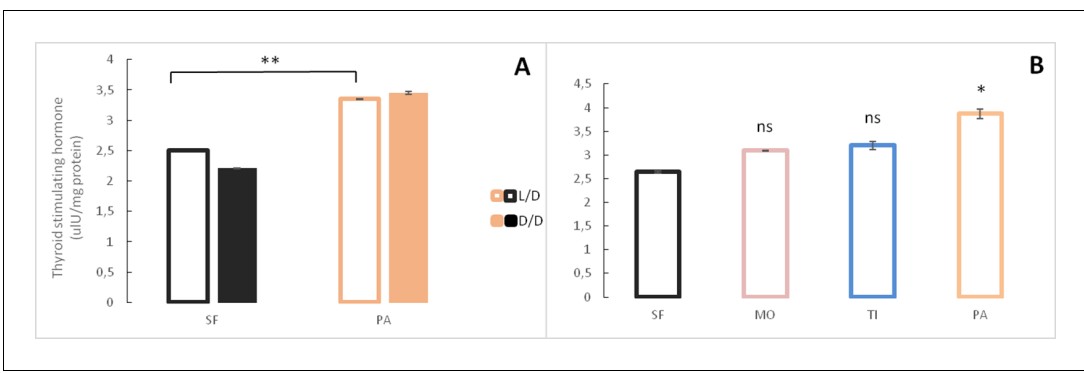

**Figure 8.** Levels of Thyroid stimulating hormone in *Astyanax mexicanus* under different experimental conditions and from different populations. (**A**) Mean Thyroid stimulating hormone levels normalized by protein concentration in adult surface fish (SF) and Pachón cavefish (PA) kept in D/D or L/D conditions for 1.5 to 2 years. N = 3 (SF L/D), 4 (SF D/D), 3 (PA L/D), 3 (PA D/D). (**B**) Mean thyroid stimulating hormone levels in SF (N = 8) and three different CF populations: Pachón (PA) (N = 5), Tinaja (TI) (N = 3) and Molino (MO) (N = 4) caves. (Error bars represent SD in three technical replicates. N ranges from 3 to 8 fish/group; *p<0.05; **p<0.01 as calculated by ANOVA and Tukey HSD Test. In B ns or * denotes significance in comparison to SF.) *Figure 8—source data 1* contains raw data and summary statistics.

The online version of this article includes the following source data for figure 8:

**Source data 1.** Raw data and summary statistics for *Figure 8*.

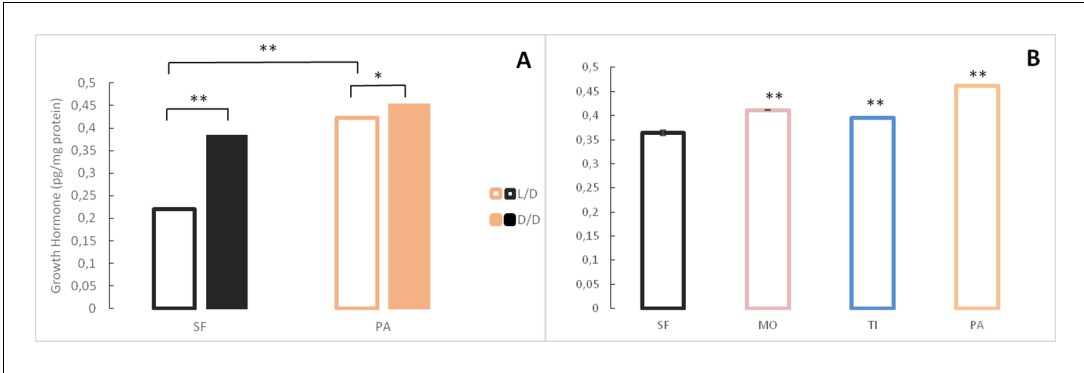

**Figure 9.** Levels of Growth hormone in *Astyanax mexicanus* under different experimental conditions and from different populations. (**A**) Mean Growth hormone levels normalized by protein concentration in adult surface fish (SF) and Pachón (PA) cavefish kept in D/D or L/D conditions for 1.5 (SF) and 2 years (PA) since < 3 dpf. N = 3 (SF L/D), 4 (SF D/D), 3 (PA L/D), 3 (PA D/D). (**B**) Mean growth hormone levels in 3–4 month old SF and three different CF populations: PA, Tinaja (TI), and Molino (MO). (Error bars represent SD in three technical replicates. N = 3 to 8/group. *p<0.05; **p<0.01 as calculated by ANOVA and Tukey HSD Test. In B, **p<0.01 compared to SF).
*Figure 9—source data 1* contains raw data and summary statistics.
The online version of this article includes the following source data for figure 9:

**Source data 1.** Raw data and summary statistics for *Figure 9*.

---

from D/D parents showed slightly enhanced plasticity and a minor shift toward lower metabolic rate and higher starvation resistance (although not statistically significant) than larvae raised from L/D parents. These results suggest that the plastic changes that appeared in the dark during the first generation may be subject to refinement during darkness in the second generation.

## Discussion

In addition to the widely accepted view that cave-adaptations result from long-term genetic processes (*Barr, 1968*; *Culver, 1982*; *Juan et al., 2010*), our results indicate that some cave-related traits can appear within a single generation by phenotypic plasticity. Exposure to constant darkness can trigger rapid metabolic, neurological, morphological and molecular changes necessary for survival of *A. mexicanus* SF colonizers, which may have allowed for the adaptations evident in extant CF. The initial plastic responses can be adaptive or non-adaptive. Subsequently, in the case of an adaptive response, plasticity would be further selected for, or in the case of non-adaptive or mal-adaptive responses, plasticity would be selected against. Our findings strongly implicate phenotypic plasticity as an important mechanism of cave colonization and rapid evolution of cave-related traits and open the possibility that genetic assimilation may be an underlying mechanism of adaptive evolution in *A. mexicanus* cavefish.

Phenotypic plasticity is often the first response of organisms exposed to an environmental change. The ability to quickly respond to novel environmental conditions is essential for initial survival and, eventually, for adaptation to occur on the genetic level (*Fox et al., 2019*; *Lande, 2009*; *Morris, 2014*). Darkness is a hallmark of the subterranean realm, the initial stress new cave colonizers encounter, and the only environmental component shared by all underground habitats (*Pipan and Culver, 2012*). Building on the work of *Rohner et al. (2013)*, we hypothesized that stress caused by perpetual darkness may be an important facilitator of the rapid appearance of novel and beneficial traits that enabled ancestral SF to endure the environmental shift associated with their entry and ultimately their adaptation to life in caves. To test this hypothesis, we used contemporary *A. mexicanus* SF as a proxy of the ancestral SF colonizers, and exposed them to complete darkness to simulate the most important change that the original SF ancestors of CF experienced when colonizing caves. Indeed, we found an increase in cortisol level in D/D exposed SF supporting our initial premise of darkness activating a stress response.

We expected to find plastic changes in D/D reared SF involving pituitary and thyroid hormonal regulation, adipose tissue, and body shape based on the previous work on dark-raised *A. mexicanus*

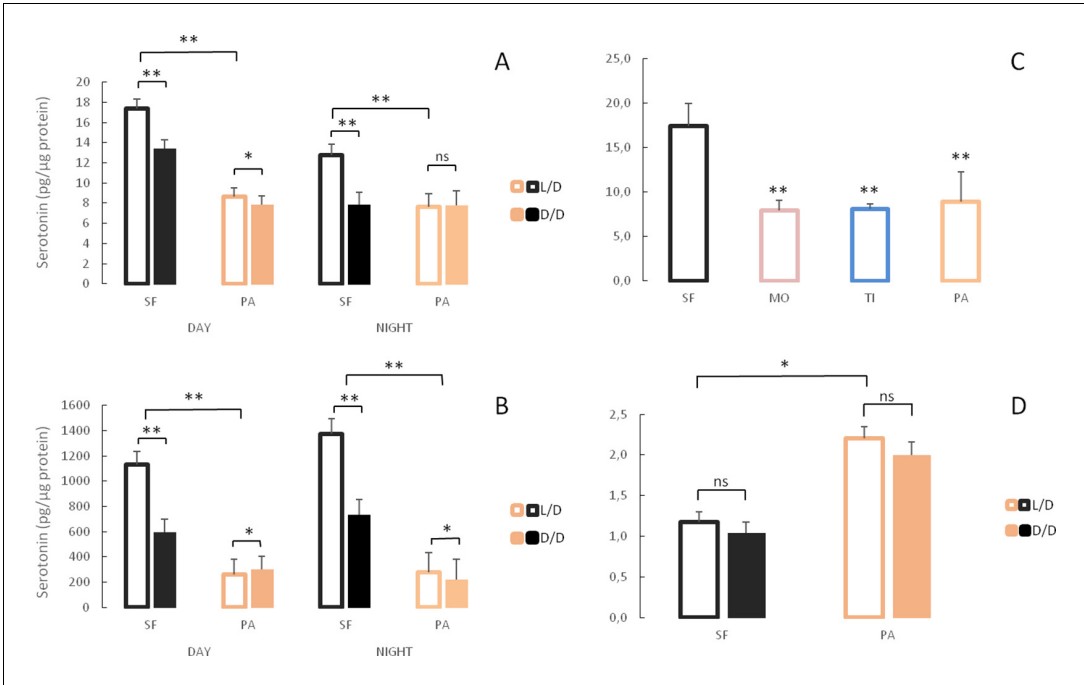

**Figure 10.** Serotonergic system changes in adults and larvae of light/dark (L/D)- and dark/dark (D/D)-reared surface fish and cavefish. (**A, B**) Serotonin levels in adult brains (A) and bodies (B) of D/D and L/D reared surface fish (SF) and Pachón cavefish (PA) collected in the middle of the day (DAY) and the middle of the night (NIGHT). (Error bars represent the standard error of the means.) (**C**) Mean serotonin levels in brains of adult SF and three different CF populations: Molino (MO), Tinaja (TI) and PA. (**D**) Mean serotonin levels in pooled samples of 5 larvae aged seven dpf placed in the experiment within first few hours post fertilization. (Error bars SEM; ns – not significant, *p<0.05; **p<0.01 as calculated by ANOVA and post-hoc Tukey HSD Test. In C, **p<0.01 vs SF. The number of each fish type subjected to analysis ranged from 4 to 10 per group.) *Figure 10—source data 1* contains raw data and summary statistics.

The online version of this article includes the following source data and figure supplement(s) for figure 10:

**Source data 1.** Raw data and summary statistics for *Figure 10*.

**Figure supplement 1.** Levels of 5-Hydroxyindoleacetic acid (5-HIAA), the main metabolite of serotonin, in adult brains of L/D or D/D-reared surface fish (SF) and Pachón cavefish (PA) collected in the middle of the day (DAY) and the middle of the night (NIGHT).

SF, Chica CF, and Los Sabinos CF (*Rasquin, 1949*). Furthermore, experiments on *Poecillia mexicana*, another teleost with cave adapted lineages, showed that darkness caused degenerate changes in the spine and promoted sexual isolation between adjacent surface and cave lineages (*Riesch et al., 2016*; *Riesch et al., 2011*; *Torres-Dowdall et al., 2018*). Although our study was unable to replicate the changes in body shape seen by *Rasquin (1949)*, we confirmed her observation of increased body fat. We also confirmed changes in pituitary and thyroid hormone metabolism, both in gene expression and hormone levels. Down-regulation of *sst1.2*, a homolog of somatostatin - growth hormone-inhibiting hormone, is consistent with increased growth hormone in D/D SF. In addition, the changes in thyroid stimulating hormone and growth hormone levels we observed in dark raised SF are consistent with the reduction of basophils and the predominance of acidophils in the pituitary gland of D/D SF reported previously (*Rasquin, 1949*). These results suggest that one of the immediate responses to cave colonization and darkness may be changes in the endocrine system. In fact, endocrine signaling, along with epigenetic changes, activation of heat shock proteins, and transcriptional regulation, are the major molecular mechanisms involved in regulating plastic responses (*Aubin-Horth and Renn, 2009*; *Beldade et al., 2011*; *Kelly et al., 2012*). Our transcriptome analyses revealed that genes and GO terms involved in all of the above processes showed significant changes in response to darkness. In addition to hormonal signaling, there are changes in genes involved in the stress response, several different signal transduction pathways, protein metabolism,

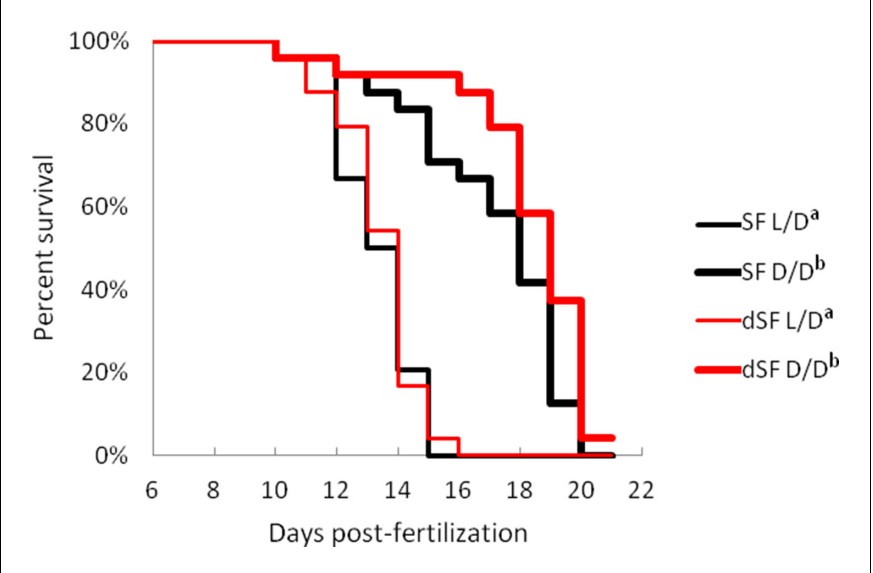

**Figure 11.** Survival curve of starvation resistance in G1 offspring of surface fish kept in normal light/dark photoperiod (SF) and G1 offspring of surface fish raised in total darkness for 2 years (dSF). Graphs show the percent of surviving fish (from the initial 24) on each day. One group of larvae from each fish type (SF, dSF) and each lighting condition (D/D, L/D) was starved starting at seven dpf (a vs. b p<0.0001). Vertical drops represent individuals lost at a given time point, groups in the legend that share a superscript are not statistically different, p values calculated by Cox proportional hazards model followed by generalized linear hypothesis test. *Figure 11— source data 1* contains raw data.
The online version of this article includes the following source data for figure 11:

**Source data 1.** Raw data for *Figure 11*.

including protein folding and posttranslational modifications, and genes showing transcription factor or epigenetic regulator activities. Furthermore, using RT-PCR we found that the expression of *hsp90aa1.2*, *dnmt1* and *dnmt3bb*, genes related to stress or epigenetic processes (*Gore et al., 2018*), were changed in D/D SF. These results suggest that multiple molecular mechanisms underlying phenotypic plasticity may be mobilized by exposure of SF to darkness.

Our RNAseq results contained functional categories enriched in both the up- and the down-regulated dataset: hormone activity (as mentioned above) and oxidation-reduction processes. Changes in oxidation-reduction processes may be related to increased stress following the exposure to a changed environment. Alternatively, these changes may be light-dependent since light activates the production of reactive oxygen species (ROS), which, in turn, act as messengers between photoreception and the circadian clock (*Hirayama et al., 2007*; *Pagano et al., 2018*). Pathways involved in the response to oxidative stress, photoreception, and circadian regulation were down-regulated. Thus, the absence of light may directly reduce ROS production, photoreception, and circadian control.

Our study revealed many traits that change in SF upon exposure to darkness. Strikingly, all these traits are associated with the cave lifestyle, and almost all of them change to resemble CF adaptive phenotypes. For example, compared to SF, CF have higher starvation resistance (*Aspiras et al., 2015*), a lower metabolic rate (*Hüppop, 1986*; *Moran et al., 2014*), higher basal cortisol levels (*Gallo and Jeffery, 2012*), and lower serotonin levels (but see *Elipot et al., 2014*). When exposed to darkness, in addition to mimicking higher growth hormone and triglyceride levels in CF (see above), we found that SF show higher starvation resistance, lower metabolic rates, higher cortisol levels, and lower serotonin levels. The results demonstrate that for these traits SF begin to resemble CF in the same generation following exposure to complete darkness. Because the plastic phenotypes are under genetic control in multiple independently evolved CF populations and irreversible when CF are transferred to light in the laboratory (*Gross, 2012*; *Jeffery, 2001*), the results are consistent with the possibility that the initial dark-induced plastic responses in SF became fixed in CF-like descendants via genetic assimilation. However, not all CF traits could be replicated by exposing

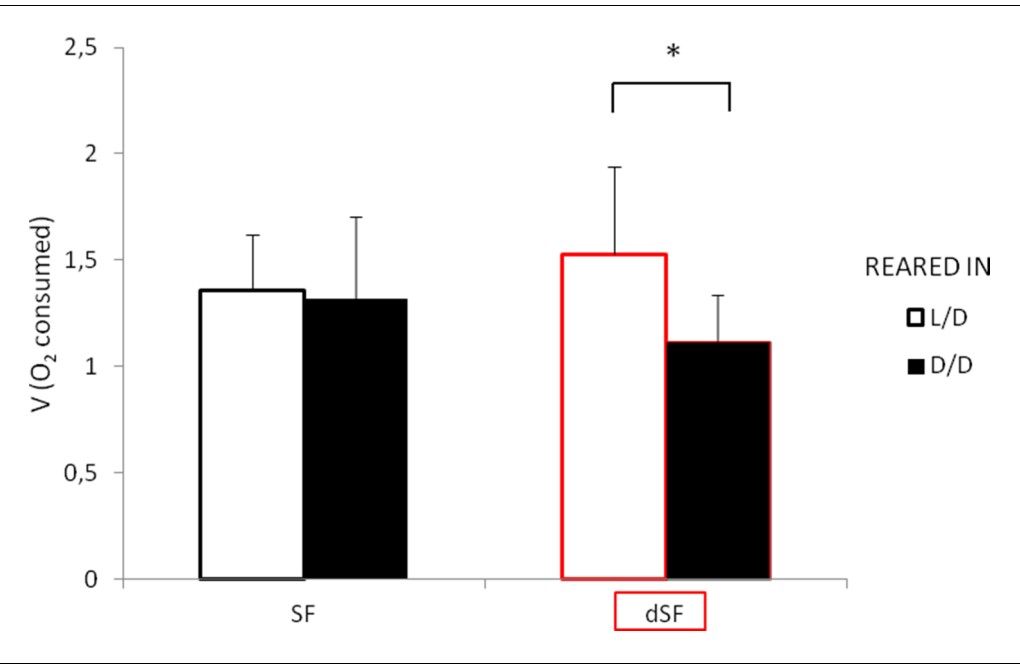

**Figure 12.** Average oxygen consumption of 11 dpf G1 offspring surface fish kept in the normal light/dark photoperiod (SF) and surface fish kept in total darkness for 2 years (dSF). Each group of offspring was exposed to D/D or L/D conditions within first 24 hpf. (Error bars represent standard deviation; *p<0.05, as calculated by ANOVA and Tukey HSD Test.). *Figure 12—source data 1* contains raw data and statistics.
The online version of this article includes the following source data for figure 12:

**Source data 1.** Raw data and statistics for *Figure 12*.

SF to darkness. Therefore, although important, plasticity caused by constant darkness is not the only path for the appearance of cavefish traits. Other mechanisms, including plasticity caused by environmental factors other than darkness (e.g. changes in nutrient availability), selection on the standing genetic variation, new mutations, or any combination of these processes, might also be involved. In addition, standing genetic variation for differences in plasticity in an ancestral population could result in the evolution of these traits.

Although darkness is an ecologically relevant cue, the induction of changes in SF by the absence of light does not necessarily imply that they are adaptive for cave life. Plasticity can result in adaptive, non-adaptive as well as maladaptive responses (*Ghalambor et al., 2007*; *Langerhans and DeWitt, 2002*). It is self-explanatory how increased triglyceride content, starvation resistance or decreased metabolic rate can increase fitness in the (food-limited) cave environment, but the effects of neurotransmitter or hormone changes are less intuitive. Therefore, we hypothesized that, if a trait is adaptive, it would be present in multiple independently evolved cavefish populations (*Prevorcnik et al., 2004*). As a proof of principle, higher starvation resistance and triglyceride content (*Aspiras et al., 2015*) conform to this criterion. We tested neurotransmitter and hormone levels and showed that growth hormone and serotonin change in multiple CF concordantly to the initial SF plastic response. Conversely, thyroid stimulating hormone showed the opposite direction of change, suggesting that the initial SF plastic response was not adaptive.

Also opposite to the expected direction of change, melanophore numbers increase in D/D SF. The increase of melanophores in dark-raised SF was modest and confined to specific regions of the body, but consistent with up-regulation of some melanosome-related genes in the transcriptome. This result was unexpected but considered reliable, since SF x CF hybrids can also show hyper-melanization (*Gross et al., 2016*), and neutral mutations are a likely cause of melanophore reduction in *A. mexicanus* CF (*Protas et al., 2007*). We found a surprising result in another classic troglomorphic trait, eye degeneration: although visual perception was a GO term associated with down-regulated genes in D/D SF transcriptome, we were unable to detect changes in eye size. However, D/D SF exhibited well-developed retinal layers, and we even recorded thickening of the inner and outer

nuclear layer, and the photoreceptor layer. These regions are cell proliferation and intercalation centers so it is possible that darkness activates mitotic activity in these retinal layers as a compensation for defective transmission of light from the lens and induced retinal stress (*Alunni et al., 2007*). The finding of increased eye size under our experimental conditions of complete darkness is puzzling, but has also been observed in a natural *Astyanax* population that has recently colonized a cave in Texas (*McGaugh et al., 2019*). Our results are consistent with studies showing that teleosts change body pigmentation and retinal phenotypes in response to light levels (*Epp, 1972*; *Kay et al., 2001*; *Tarboush et al., 2016*). For example, *Astyanax* F2 hybrids with small or absent eyes also show hyperpigmentation, which has been explained as background adaptation due to the lack of visual input (*Gross et al., 2016*). Overall, our results suggest that plasticity lacks the propensity to produce adaptive variants, and non-adaptive, even maladaptive, effects seem to be as likely as adaptive outcomes.

Importantly, pigmentation and eyes are absent in PA and other fully troglomorphic CF. The fact that they showed non- or mal-adaptive plastic response in SF exposed to darkness, suggests that natural selection acting against plasticity may be involved in the evolution of these traits in CF. A novel aspect of the evolution of classical troglomorphic traits, the reduction of eyes and pigmentation, may be that they are a consequence of strong selection against non- or mal-adaptive plastic responses during colonization of caves by surface ancestors.

To test if the inheritance of some of the plastic phenotypic changes described here occur during the next generation, we induced spawning in D/D SF and investigated starvation resistance and metabolic rate in the offspring (dSF), traits shown to exhibit plasticity in the parental D/D generation. Neither of these traits showed differences between offspring spawned in the light or dark when maintained in L/D. However, when maintained in D/D, dSF showed continued plasticity of both traits, as well as potential adaptive-tuning of the character states, suggesting that plasticity is evolvable and can itself be an evolutionary adaptation. Increased plasticity was previously suggested to be an important stepping stone in the process of adaptations to extreme environments (*Lande, 2015*; *Lande, 2009*). Therefore, consistent with the Baldwin effect (*Crispo, 2007*), the evolution of cave related traits may have proceeded through initial selection for continued plasticity and refinement of the characters, followed by selection of individuals carrying the most improved variants of the phenotype. Higher growth hormone and triglyceride levels, as well as lower body serotonin levels in PA than D/D SF, also imply refinement of the initial SF plastic response in CF. In nature SF are periodically swept into caves and plasticity in many traits that enable survival may be critical to extend their lifespan long enough to leave offspring. According to our results, the next generation may be slightly better equipped for coping with the cave environment by increased plasticity and small adaptive switches in traits.

One of the reasons the role of plasticity in adaptive evolution has been contested is its randomness in the production of outcomes where non-adaptive and mal-adaptive responses are as likely as adaptive responses. However, our study suggests that even the initial non-adaptive plastic responses may eventually produce an adaptive outcome. Some traits, including the hallmarks of troglomorphic adaptations (the loss of eyes and pigmentation) initially show non-adaptive plastic responses but the complete loss of these traits in CF indicates strong selection against plasticity in SF colonizers. Non-adaptive plasticity may have a greater role in evolution than previously appreciated because it may increase the strength of selection against plasticity, and perhaps, in some cases, may affect the whole trait itself. Conversely, in cases where plasticity produces adaptive outcomes, initial selection will act to enhance plasticity, and plasticity will be maintained in the derived lineage for many generations. Overall, plasticity has a critical role in producing variability (adaptive or not) on which selection can act (for or against) to produce individuals with optimal fitness in the new environment.

We also uncovered two cases in which the plasticity of a single trait changes within an individual over the course of development. Both metabolic rate and serotonin showed shifts in plasticity depending on developmental stage. Metabolic rate initially (at 2.5 dpf) changes in response to the light regime but later (by 7.5 dpf) this response is lost in PA. In contrast, serotonin levels are plastic in adults but not in 7 dpf larvae. Interestingly, switches in plasticity occur at the same time as switches in character state. At 2.5 dpf there is no difference in metabolic rate between SF and PA, but by 7.5 dpf, the metabolic rate in PA is lower than in SF, agreeing with previous studies done on adults (*Hüppop, 1986*; *Moran et al., 2014*). In the case of serotonin, character states show opposite relationships at different stages: they are lower in larvae but higher in adult SF compared to PA.

Therefore, for metabolic rate and serotonin levels, both the trait and its plasticity change concordantly, suggesting that the underlying mechanisms of character development may also influence the ability to respond to environmental stimuli. These two examples show that plasticity is not an intrinsically fixed property of a trait but can develop by itself. In our examples, plasticity develops concordantly with the traits it affects, and it can develop in a direction of expansion or reduction. The development of metabolic rate in PA is accompanied by canalization of the same trait. In the case of serotonin, development of the 5-HT system is accompanied by an increase in its plasticity.

Phenotypic plasticity can fill several gaps in the current model of cave colonization and the evolution of cave dwelling organisms. Our study shows how a colonizing surface ancestor, having no adaptations to caves and presented with the challenges of low food availability, orientation, and reproduction in complete darkness, could have overcome a transition to the extreme subterranean environment and establish a cave-adapted lineage. Further, due to plasticity, numerous adaptive traits arise within a single generation in response to only one, albeit the most relevant, environmental cue: darkness. Finally, plasticity may enable the maintenance of distinct phenotypes in the face of gene flow from ancestral surface-dwelling populations. The incorporation of phenotypic plasticity as one of the mechanisms underlying cave colonization and adaptive evolution provides a solution to dilemmas associated with drift or selection (*Cartwright et al., 2017*) acting alone to induce cave-related phenotypes in *A. mexicanus*. Because multiple successful transitions by the same surface ancestors are not unique to *Astyanax* (*Carlini et al., 2009*; *Verovnik et al., 2004*), some animal groups have colonized caves repeatedly in different times and geographical regions, and several unique cave dwellers are closely related to highly invasive (and plastic) surface dwelling species (*Bilandžija et al., 2013*; *Kupriyanova et al., 2009*), it is possible that plasticity may be a general phenomenon in the colonization and adaptation to cave environments.

# Materials and methods

## Key resources table

| Reagent type (species) or resource | Designation | Source or reference | Identifiers | Additional information |
|---|---|---|---|---|
| Biological sample (Astyanax *mexicanus* surface fish) | Surface fish, SF | Jeffery laboratory | | |
| Biological sample (Astyanax *mexicanus* Pachón cavefish) | Pachón, cavefish, CF, PA | Jeffery laboratory | | |
| Biological sample (Astyanax *mexicanus* Tinaja cavefish) | Tinaja, TI | Jeffery laboratory | | |
| Biological sample (Astyanax *mexicanus* Molino cavefish) | Molino, MO | Jeffery laboratory | | |
| Commercial assay or kit | TruSeq mRNA Library Prep Kit | Illumina | Cat# RS-122–2001 | |
| Commercial assay or kit | Cortisol ELISA Kit | Cayman Chemical | Cat#500360 | |
| Commercial assay or kit | Pierce BCA Protein Assay Kit | Thermo Fisher Scientific | Cat#23225 | |
| Commercial assay or kit | Fish growth hormone(GH) ELISA Kit | Cusabio | Cat#CSB-E12121Fh | |
| Commercial assay or kit | Fish thyroid stimulating hormone(TSH) ELISA Kit | Cusabio | Cat#CSB-EQ02726Fl | |
| Commercial assay or kit | Triglyceride Quantification Assay Kit | Abcam | Cat#ab65336 | |
| Chemical compound, drug | Tricaine methanesulfonate | Western Chemical Inc | Cat#TRS1 | |

*Continued on next page*

*Continued*

| Reagent type (species) or resource | Designation | Source or reference | Identifiers | Additional information |
|---|---|---|---|---|
| Chemical compound, drug | Trizol | Invitrogen | Cat#15596026 | |
| Chemical compound, drug | Superscript III and IV Reverse Transcriptase | Invitrogen | Cat#18080044, 18090050 | |
| Chemical compound, drug | NP-40 | Abcam | Cat#ab142227 | |
| Software, algorithm | ImageJ | https://imagej.nih.gov/ij/ | RRID:SCR_003070 | |
| Software, algorithm | FASTQC | http://www.bioinformatics.babraham.ac.uk/projects/fastqc/ | RRID:SCR_014583 | |
| Software, algorithm | Bowtie2 v2.3.2 | http://bowtie-bio.sourceforge.net/bowtie2/index.shtml | RRID:SCR_005476 | |
| Software, algorithm | TopHat v2.1.1 | https://ccb.jhu.edu/software/tophat/index.shtml | RRID:SCR_013035 | |
| Software, algorithm | Cufflinks v2.1.1 | http://cole-trapnell-lab.github.io/cufflinks/ | RRID:SCR_014597 | |
| Software, algorithm | R package 'cummeRbund' | https://bioconductor.org/packages/release/bioc/html/cummeRbund.html | | |
| Software, algorithm | R.Studio v1.0.136 | https://www.rstudio.com/products/rstudio/#Desktop | | |
| Software, algorithm | R v3.5.1 | https://cran.r-project.org/bin/windows/base/old/3.5.1/ | | |
| Software, algorithm | DAVID Bioinformatics Resources | https://david.ncifcrf.gov/ | RRID:SCR_001881 | |
| Software, algorithm | Reactome | https://reactome.org/ | RRID:SCR_003485 | |
| Software, algorithm | RefFinder | https://www.heartcure.com.au/for-researchers/ | RRID:SCR_000472 | |
| Software, algorithm | R version 3.5.3 | https://cran.r-project.org/src/base/R-3/ | | |
| Software, algorithm | R package 'multcomp' | https://cran.r-project.org/web/packages/multcomp/index.html | | |
| Software, algorithm | CSW32 data program | https://www.dataapex.com/products/csw32.php (product was discontinued) | | |
| Software, algorithm | SigmaStat version 3.5 | https://sigmastat.software.informer.com/3.5/ | RRID:SCR_010285 | |

## Animals and experimental conditions

The study groups included *Astyanax mexicanus* SF and CF primarily from Pachón Cave (PA). In some experiments, CF from Tinaja (TI) and Molino (MO) caves were also used. Animals were obtained from the Jeffery Laboratory colony at the University of Maryland, and had undergone 3–5 generations in captivity since their original collection from the caves in 2002 (PA, TI) or 2006 (MO). Fish in the main colony were kept in 40 L tanks in cohorts of 8–15 animals and exposed to a 14 hr light/10 hr dark photoperiod. Spawning was induced every two weeks by extra feeding a few days prior to increasing the water temperature (*Jeffery et al., 2000*). Temperature regime changes were as follows: fish were normally kept in 22 ˚C, on the first day of spawning temperature was raised to 24 ˚C, on the second day raised to 26 ˚C, on the third day returned to 24 ˚C, and on the fourth day returned to 22 ˚C. Embryos were spawned in the night and collected in the morning by washing them from the breeding nets. Dead embryos were removed and the remainder transferred to clean fish system water containing methylene blue. The fish were raised and handled according to established University of Maryland and NIH guidelines and all experiments conform to the regulatory standards.

Fish embryos spawned at the same time from different parent tanks were mixed in order to minimize the effects of specific genetic backgrounds and mimic the situation in nature where fish are randomly swept into caves. Mixed embryo clutches were divided in half and randomly assigned to different experimental conditions. One group was initially exposed to darkness either as embryos or young fry (from 1 to 2 hr to 3 days after spawning) and then raised in complete darkness (D/D) for up to two years. The other group, originating from the same brood, was reared in a normal 14 hr light/10 hr dark photoperiod (L/D). All other conditions were identical between the D/D and L/D study groups. Fish were reared in plastic tanks with no filtration or running water. Depending on their size and age, fish were reared in 1, 3 or seven liter tanks and then transferred to larger tanks as they increased in size. Once the fish reached approximately 1 cm in length they were transferred to 3L tanks, and once they grew to over 2 cm they were transferred to 7L tanks. Densities of fish among different groups depended on survival but were approximately similar and ranged between 5 to 10 fish per tank. Water was changed approximately once a month at the same time for every tank in the experiment. Fish in each tank were fed the same volume of brine shrimp once per day, approximately 0.5 mL of concentrated brine shrimp was added to smaller fish in 1L tanks, 1 mL in larger 3L tanks, and 2 mL in the largest 7L tanks. All procedures followed standard procedures for *Astyanax* fish husbandry in the Jeffery Laboratory. Dim red light (25 watts) was used when it was necessary to feed or handle the fish in the dark (*Romero, 1985*). Dead fish were removed when they were observed. Fish of different ages were used throughout the experiments and were all adults (unless otherwise noted). Within every experiment fish groups of comparable ages were used for comparisons. Fish from both sexes were included in each experiment, and fish used in each experiment were collected at the same time of the same day.

For the G1 generation experiments, SF placed in D/D as embryos and raised for 2 years were spawned using standard procedures as explained above. The spawned embryos (dSF) were divided in two groups within the first 24 hr post-fertilization (hpf), and one group was left in D/D conditions and the other group was placed in L/D photoperiod within first 24 hpf. A control group of SF spawned from L/D raised parents was also divided into two groups and raised in L/D or D/D during the period of experimentation.

Information on sample sizes, time lapsed in experimental conditions, and the ages of fish used in specific experiments is provided in the sections corresponding to the description of specific experimental methods.

The following procedure was used for collecting, and when necessary pulverizing, adult fish for the respective experiments: fish were sacrificed with 0.4 or 0.5 mg/L MS222, tricaine methanesulfonate (Western Chemical Inc, Ferndale, WA, USA). Fish were maintained in the dark until they expired and then photographed, weighed, and pulverized in liquid nitrogen using a mortar and pestle. The pulverized material was divided into several Eppendorf tubes and stored at −80 ˚C until further processing. For the brain chemistry experiment, whole brains were rapidly removed, placed in 1.5 mL Eppendorf tubes, immediately frozen on dry ice and stored at −80 ˚C until analysis.

## Morphological analysis

After photography, the body length (total: rostrum to tail tip, fork: rostrum to tail fork, and standard: rostrum to beginning of the tail), the dorsoventral width of the body (at the level of the operculum and at the beginning of the dorsal fin), the eye diameter, and the pupil diameter of 16 D/D fish and 15 L/D from each experimental condition were measured using ImageJ software.

Five fish from each group, kept under the experimental conditions described above for 9 months beginning two hpf, were used for melanophore quantification. Fish were fixed for 1 hr in 4% paraformaldehyde, washed three times in PBS, and examined under a stereomicroscope. Pigment cells were counted as in *Bilandžija et al. (2018)* in four different places on the left side of each fish: in the proximal tail fin stripe, below the dorsal fin, along the dorsal flank of the trunk, and on the chin. Eyes of four SF kept in D/D and three L/D controls kept in experimental conditions for 1.5–2.5 years were embedded in paraplast blocks, sectioned, stained, and morphometric analyses was performed on retinal layers as described previously (*O'Quin et al., 2013*). To account for intra-retinal variation caused by changes in location or sectioning, we measured retinas in the middle of the region between the optic nerve and the ora serrata on fifteen test fields randomly chosen across five sections for every fish. In order to compare these retinal measurements among individual fish for

statistical analysis, we used eye diameters as independent parameters for normalization of our data (*Collery et al., 2014*).

## RNA sequencing, differential expression analysis, and identification of enriched pathways

Three surface fish kept in D/D and three in L/D experimental conditions for 7 months post-fertilization were used for RNA sequencing. Prior to RNA extraction, the stomach and liver were removed from the fish because they compromised RNA quality. Total RNA was isolated from samples stored in TRIzol reagent according to the manufacturer's instructions (Invitrogen, Carlsbad, CA, USA). The fragment size, concentration, RNA integrity number (RIN) and 28S/18S ratio of RNA extracts were determined using an Agilent 2100 Bioanalyzer (Agilent, Santa Clara, CA, USA). The cDNA libraries were constructed using TruSeq mRNA Library Prep Kit (Illumina, San Diego, CA, USA). Fragment sizes and concentrations of libraries were verified using an Agilent 2100 Bioanalyzer and ABI StepOnePlus Real-Time PCR machine (Thermo Fisher Scientific, Waltham, MA, USA). The samples were sequenced in a single Hiseq X-ten lane with a strategy of 150 bp paired-ends (PE), resulting in about 8 Gb raw data for each sample. To avoid potential adapter contamination, the last 60 bp of all reads were trimmed. Subsequently, the reads were removed if they contained: 1) more than 50% low quality (Q < 35) bases; 2) more than 10 Ns, and 3) were PCR duplications. Cleaned read quality was confirmed in FASTQC (*Andrews, 2010*), and a total of 230,437,086 cleaned reads were retained for transcriptomic analysis. The reads have been submitted as bioproject PRJNA557727 (accession numbers SRX6631237 - SRX6631242).

A local database was built on the Carbonate Cluster implemented by the National Center for Genome Analysis Support (*Stewart et al., 2017*) using the *Astyanax mexicanus* draft genome v102.93 (GCA_000372685.1) and associated gene model annotation files. The index of the reference genome was built using Bowtie2 v2.3.2 (*Langmead and Salzberg, 2012*), and paired-end clean reads were aligned to the reference genome using TopHat v2.1.1, with max-intron-length set to 1000 (*Kim et al., 2013*). Transcript coverage and differential expression were determined following the Cufflinks pipeline (*Trapnell et al., 2010*). Briefly, mapped reads for each sample were assembled using Cufflinks v2.1.1 with the parameters `–frag-bias-correct`, `–multi-read-correct`, `–upper-quartile-norm`, and `–compatible-hits-norm`. Transcripts were merged using default Cuffmerge parameters, and Cuffquant with `–frag-bias-correct` and `–multi-read-correct` was implemented to quantify transcript expression. Significant changes between transcripts were calculated using Cuffdiff (*Trapnell et al., 2012*) with parameters `–compatible-hits-norm`, `–frag-bias-correct`, and `–multi-read-correct` for light and dark treatments. Results were visualized using the package CummeRbund (*Goff et al., 2014*) implemented in R.Studio v1.0.136 (*R Studio T, 2015*) using R v3.5.1 (*R Development Core Team, 2018*).

Significantly differentially expressed genes ($p_{adj}$ < 0.1) were subjected to Gene Ontology (GO) and Kyoto Encyclopedia of Genes and Genomes (KEGG) pathway enrichment analysis as implemented on DAVID Bioinformatics Resources (*Huang et al., 2009a*; *Huang et al., 2009b*) and pathways were investigated more closely using Reactome (*Fabregat et al., 2016*).

## Real time quantitative RT-PCR

Total RNA was isolated from 10 to 12 month old adult SF and PA placed in D/D or L/D conditions within the first few hours after fertilization. After the liver and stomach were removed, the RNA from the remaining body was isolated with Trizol Reagent (Thermo Fisher Scientific, Waltham, MA, USA) according to the manufacturer's instructions and treated with DNase to remove genomic DNA if present. Equal amounts of RNA (1.5 µg) were reverse transcribed using Superscript III or IV (Invitrogen, NY, USA). The cDNA concentration was adjusted to 12.5 ng/µL and up to 50 ng were used in each reaction. We tested 12 different HKG (housekeeping genes): (*ube2a*, *rpl13a*, *eef1a1l1*, *nek7*, *tbcb*, *rnf7*, *rpl27*, *ndufa6*, *ap5s1*, *mob4*, *lsm12a*, *actc1b*) using RefFinder, which is located on the Cotton EST Database webpage (*Xie et al., 2011*), and integrates currently available major computational programs (geNorm, Normfinder, BestKeeper, and the comparative ΔΔCt method) to compare and rank the candidate reference genes. The geometric mean of the three most stable HKG (*tbcb*, *mob4* and *rnf7*) was used for normalization. Primers for genes of interest were designed across exon/intron boundaries except for *pdx1*, where this was not possible (Supplementary File 1). Primer

efficiencies were calculated in Excel from standard curves, and those with efficiencies between 90% and 110% were used for quantifications. The resulting PCR products were sequenced to confirm the targeted genes. Relative expression was calculated according to a formula $\Delta Ct = E^{\wedge(Ct\ reference\ -\ Ct\ target)}$ (E = 1+primer efficiency/100), which enables the comparison of normalized expression of a gene between multiple samples, instead of ratios between two samples, taking primer efficiency into account. Statistical significance of observed expression differences was calculated by ANOVA and adjusted by Bonferroni index.

## Starvation resistance determinations

SF and PA embryos were placed in L/D vs. D/D experimental conditions within 24 hpf. At 7 dpf, the first day larval fish were fed brine shrimp, a group of 36 larvae from L/D and 36 from D/D per fish type was fed. Groups of 36 larvae from each experimental condition and each fish type were not fed. The fish were identically treated in all other aspects, and the number of living fish was recorded daily. Dead fish were removed each day. The experiment was repeated 3 times independently.

Statistical analyses for survival analysis were carried out using R version 3.5.3 (*R Development Core Team, 2018*). Differences in starvation resistance were compared with a Cox proportional hazards model using the function coxph() from the R package 'survival' (*Therneau and Grambsch, 2000*). For the parental generation, the model included the covariates fish (PA vs. SF), light status (D/D vs. L/D), and fish x light status. For the G1 generation, the model included the covariates fish (dSF vs. SF), light status (D/D vs. L/D) and fish x light status. If the null hypothesis that all β = 0 was rejected, *post-hoc* pairwise comparisons of contrasts were carried out using the function `glht()` from the package multcomp with the specification mcp = 'Tukey' (*Hothorn et al., 2008*). Proportionality assumptions for Cox proportional hazards models were tested using the `coxphz` function (*Therneau and Grambsch, 2000*).

## Metabolic rate determinations

Unhatched SF and PA embryos or 7 dpf larvae were placed individually in air-tight glass vials filled to the top with fish system water that contained no air bubbles. Half of the vials from each type were kept in L/D and the other half were wrapped in aluminum foil and kept in the same way in D/D. After two days, the $O_2$ remaining in each vial was measured using a Membrane Inlet Mass Spectrometer (MIMS) Machine (Bay Instruments, Easton, MD, USA). The amount of $O_2$ consumed per fish was calculated by subtracting the measured amount of $O_2$ in the vials with fish from the mean amount of $O_2$ measured in blank vials without fish. This experiment was repeated twice with embryos and once with larvae. The same methods were used in all replications.

## Cortisol quantification

We quantified cortisol from adult SF and PA that were maintained in D/D or L/D experimental conditions for 1.5 to 2 years beginning before 3 dpf. Cortisol was extracted using previously developed procedures (*Canavello et al., 2011*; *Gallo and Jeffery, 2012*). Briefly, tissue samples stored at −80 ° C were thawed and homogenized in PBS after which 5 mL of diethyl ether was added. Following centrifugation for 5 min at 3500 x g, the cortisol-containing top layer was removed and evaporated in a fume hood overnight. The cortisol was reconstituted in PBS overnight at 4 °C and quantified using the Cortisol ELISA Kit (Item № 500360, Cayman Chemical, Ann Arbor MI, USA) following the kit protocol. Cortisol samples from fish within each group were pooled and assayed in triplicate. The recorded absorbance was compared to a standard curve to determine the cortisol concentration, and each sample was standardized to protein concentration determined by Pierce BCA Protein Assay Kit (Thermo Fisher Scientific, Waltham, MA, USA). The experiment was repeated 3 times on independent batches of fish.

## Triglyceride quantifications

Pulverized fish were weighed and 20 µl per mg of 6% NP-40 (Abcam, Cambridge, UK) was added to each sample. Samples were heated to 80–100℃ for 2–5 min, and the process was repeated 2–3 times. Insoluble material was removed by centrifugation for 2 min at 13,000 rpm, and equal volumes of supernatants from all samples of the same fish type were pooled and diluted 20x in $ddH_2O$ before ELISA. For ELISA, 50 µL of each sample was assayed in triplicate using the Triglyceride Quantification

Assay Kit (ab65336), (Abcam, Cambridge, UK) according to the manufacturer's instructions. The procedure followed a previously published method (*Aspiras et al., 2015*). Concentrations were determined from the standard curve and standardized to protein concentration as described above. The experiment was repeated on 3 independent batches of fish.

## Hormone quantifications

Levels of growth hormone (CSB-E12121Fh) and thyroid stimulating hormone (CSB-EQ02726Fl) were quantified by ELISA using commercially available kits from Cusabio (Wuhan, China). Adult SF kept in D/D for 1.5 years and PA kept in the experiment for almost 2 years were used. For comparison of hormone levels between different populations, 3 to 4 month old SF, PA, TI, and MO were used. The fish tissue was weighed, and 3 μl PBS/mg tissue was added to each sample. After two freeze-thaw cycles and homogenization, the homogenate was centrifuged at 5000 x g for 5 min at 4 °C. Equal volumes of supernatant from each fish type were pooled and immediately loaded on the plate, 50 μl per well, in triplicate. Assays were performed following the manufacturer's instructions and concentrations were determined according to the standard curve. Results were standardized to protein concentration as described above. Each experiment was repeated 3 times independently.

## Serotonin quantification

Brains were dissected from PA and SF kept in L/D or D/D experimental conditions for 1.5 to 2 years beginning at or before 3 dpf, and from adult SF (N = 8), PA (N = 5), TI (N = 5), and MO (N = 6) kept in the main fish system and HPLC was used to quantify neurotransmitter levels as in *Bilandžija et al. (2018)*. In addition, we quantified serotonin (5-HT) in samples of pooled five larvae (7 dpf) placed in the L/D or D/D experimental conditions within first few hours after fertilization, N = 6 (SF L/D), 5 (SF D/D), 5 (PA L/D), and 4 (PA D/D). Serotonin and its metabolite, 5-hydroxyindoleacetic acid (5-HIAA), were analyzed using HPLC with electrochemical detection as described previously with minor modifications (*Renner and Luine, 1986*). Brains were placed into 100 μL of sodium acetate buffer (pH 5.0) containing the internal standard alpha-methyl dopamine (αMDA; Merck and Co., Inc, Kenilworth, NJ). Fish bodies were placed in either 200 to 400 μL of acetate buffer containing αMDA based on the amount of tissue present. Both tissue types were disrupted by sonication using a 4710 Ultrasonic Homogenizer (Cole-Parmer Instrument Co., Chicago IL) and stored at −80° C. Prior to analysis, the sonicated brain samples were thawed, 4 μL of 1 mg/mL ascorbate oxidase (Sigma-Aldrich, St. Louis, MO, USA) was added to each sample and the samples were centrifuged at 17,000 g for 15 min. Fish body samples were treated the same way except that the supernatant was centrifuged a second time through a 0.2 μm filter. The filtered supernatant was removed and a Waters Alliance e2695 separation module was used to inject 50 μL of the supernatant onto a $C_{18}$4 μm NOVA-PAK radial compression column (Waters Associates, Inc Milford, MA) held at 30°C. The initial mobile phase (pH 4.1) was prepared using 8.6 g sodium acetate, 250 mg EDTA, 14 g citric acid, 80 mg octylsulfonic acid, and 80 mL methanol in 1 L of distilled water (monoamine standards and chemicals were purchased through Sigma-Aldrich) and adjusted with small additions of octylsulfonic acid, glacial acetic acid, and methanol to optimize the separation. Electrochemical detection was accomplished using an LC four potentiostat and glassy carbon electrode (Bioanalytical Systems, West Lafayette, IN, USA) set at a sensitivity of 0.5 nA/V (brain samples) or 1 nA/V (body samples) with an applied potential of +0.7 V versus an Ag/AgCl reference electrode. The pellet was solubilized in 400 μL of 0.4 N NaOH and protein content was analyzed using the Bradford method (*Bradford, 1976*). A CSW32 data program (DataApex Ltd., Czech Republic) was used to determine 5-HT and 5-HIAA concentrations in the internal standard mode using peak heights calculated from standards. Injection versus preparation volumes were corrected and amine concentrations were normalized by dividing pg amine by μg protein. Data were tested for differences using a Three Way Analysis of Variance (SigmaStat version 3.5, Systat Software Inc, San Jose, CA). In analyses that revealed a significant effect between groups, the Holm-Sidak method was used to conduct pairwise comparisons. All data were tested for the presence of outliers using the Grubb's test (*Rohlf and Soka, 1981*). Based on this analysis two brain (PA D/D night, SF L/D day) and three body samples (PA D/D night, PA L/D day, SF D/D day) were deleted from the data set. In the end a total number of fish used in the analysis for brains was 9 (SF L/D day), 10 (SF D/D day), 10 (PA L/D day), 10 (PA D/D day), 7 (SF L/D night), 6 (SF D/D night), 4 (PA L/D night), 4 (PA D/D night) and for the bodies 10

(SF L/D day), 9 (SF D/D day), 8 (PA L/D day), 10 (PA D/D day), 7 (SF L/D night), 7 (SF D/D night), 4 (PA L/D night), 5 (PA D/D night).

## The G1 offspring of dark-raised surface fish

We measured metabolic rate and starvation resistance in the G1 offspring of dark raised (dSF) and control SF using the methods described above. Statistical analysis was also done using the methods described above with N = 24 larvae per group in starvation resistance experiments and N = 10 (L/D SF), 13 (D/D SF), 12 (L/D dSF) and 15 (D/D dSF) in metabolic rate experiments.

## Acknowledgements

We wish to thank all members of the Jeffery lab, especially Ruby Dessiatoun, for taking care of the fish.

## Additional information

### Funding

| Funder | Grant reference number | Author |
|---|---|---|
| National Eye Institute | EY024941 | William Jeffery |
| Croatian Science Foundation | Tenure Track Pilot Programme TTP-2018-07-9675 | Helena Bilandžija |
| FP7 People: Marie-Curie Actions | FP7-PEOPLE-2011-COFUND – NEWFELPRO FP7 2007-2013, grant agreement no. 291823, project EACAA, agreement No. 50 | Helena Bilandžija |
| National Science Foundation | 1556819 | Megan L Porter |
| National Science Foundation | IOS 1256898 | Kenneth J Renner |
| New International Fellowship Mobility Programme for Experienced Researchers | FP7-PEOPLE-2011-COFUND – NEWFELPRO FP7 2007-2013, grant agreement no. 291823, project EACAA, agreement No. 50 | Helena Bilandžija |
| École Polytechnique Fédérale de Lausanne | Tenure Track Pilot Programme TTP-2018-07-9675 | Helena Bilandžija |

The funders had no role in study design, data collection and interpretation, or the decision to submit the work for publication.

### Author contributions

Helena Bilandžija, Conceptualization, Data curation, Formal analysis, Supervision, Funding acquisition, Investigation, Visualization, Methodology, Project administration; Breanna Hollifield, Guanliang Meng, Mandy Ng, Romana Gračan, Investigation; Mireille Steck, Investigation, Methodology; Andrew D Koch, Formal analysis, Investigation; Helena Ćetković, Resources, Supervision; Megan L Porter, Resources, Formal analysis, Supervision, Funding acquisition; Kenneth J Renner, Resources, Formal analysis, Supervision, Investigation, Methodology; William Jeffery, Conceptualization, Resources, Supervision, Funding acquisition, Project administration

### Author ORCIDs

Helena Bilandžija (iD) https://orcid.org/0000-0002-4922-0149
Guanliang Meng (iD) https://orcid.org/0000-0002-6488-1527
Mandy Ng (iD) https://orcid.org/0000-0001-8267-8330

Andrew D Koch (iD) https://orcid.org/0000-0003-2418-2665
Romana Gračan (iD) https://orcid.org/0000-0002-7680-1993
Megan L Porter (iD) https://orcid.org/0000-0001-7985-2887
William Jeffery (iD) https://orcid.org/0000-0002-6997-2946

### Ethics

Animal experimentation: This study was performed in strict accordance with the recommendations in the Guide for the Care and Use of Laboratory Animals. All of the animals were maintained and handled according to Institutional Animal Care guidelines of the University of Maryland, College Park (IACUC #R-NOV-18-59) (Project 1241065-1). All surgery was performed under anesthesia and every effort was made to minimize suffering.

### Decision letter and Author response

Decision letter https://doi.org/10.7554/eLife.51830.sa1
Author response https://doi.org/10.7554/eLife.51830.sa2

## Additional files

### Supplementary files

• Supplementary file 1. List of genes and primers used in RT-PCR experiments.

• Supplementary file 2. Summary statistics of Illumina output: the number reads, total base pairs, quality trimmed reads retained for each treatment, and the overall mapping rate from Tophat2 using Bowtie2.

• Transparent reporting form

### Data availability

Sequencing data for transcriptome analysis have been deposited in NCBI as Bioproject PRJNA557727 All data generated during this study are included in the manuscript and supporting files. Source data files have been provided for Figures 1, 3, 4, 5, 6, 7, 8, 9, 10, 11, and 12.

The following dataset was generated:

| Author(s) | Year | Dataset title | Dataset URL | Database and Identifier |
|---|---|---|---|---|
| Bilandžija H, Steck M, Meng G, Porter ML, Jeffery WR | 2019 | Astyanax mexicanus breed:Surface Fish (Mexican tetra) | https://www.ncbi.nlm.nih.gov/bioproject/?term=PRJNA557727 | NCBI BioProject, PRJNA557727 |

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
