## [Decision Letter]

**Acceptance summary:**

Both reviewers and the editors agree that the convincing results of this study will be of interest to the broader scientific research communities. The manuscript provides key insights into the role of plasticity in evolution, and should have a broad, general impact on the fields of evolutionary biology, developmental biology, and evo-devo.

**Decision letter after peer review:**

Thank you for submitting your article "Phenotypic plasticity as a mechanism of cave colonization and adaptation" for consideration by *eLife*. Your article has been reviewed by two peer reviewers, and the evaluation has been overseen by a Reviewing Editor and Patricia Wittkopp as the Senior Editor. The reviewers have opted to remain anonymous.

The reviewers have discussed the reviews with one another and the Reviewing Editor has drafted this decision to help you prepare a revised submission.

Summary:

The paper "Phenotypic plasticity as a mechanism of cave colonization and adaptation" describes the results of raising surface fish in the dark to address the role of phenotypic plasticity in the evolution of cave animals. This work addresses a long-standing question: What happens upon initial colonization of a novel environment? Towards this end, the authors present a large amount of data, including RNA-sequencing and phenotypic data for multiple traits, which together suggest that phenotypic plasticity could play a large role in colonization in a novel, extreme environment, the cave. This work is an important first step in addressing the role of phenotypic plasticity in colonization of and evolution to a novel environment and is broadly applicable, as the principles discussed here have the potential to apply to many organisms colonizing new environments.

Essential revisions:

1) For a few traits, it appears that only one biological replicate was performed. These include cortisol levels, triglyceride levels, TSH (thyroid stimulating hormone) levels and GH (growth hormone) levels. It may be that I am misunderstanding the data. However, if this is the case, I recommend running more biological replicates. In line with this, the second generation survival results are rather mild. Can the authors provide information around the number of times this assay was performed, n values, etc.?

2) The reviewers are concerned that some of the conclusions are overstated in the Discussion. While the authors make a strong argument that many cave phenotypes can be phenocopied by phenotypic plasticity, they do not think from this study that claims can be made about natural selection acting to enhance or reduce plasticity. Specifically

A) The initial sentence of the Discussion suggests that the plastic responses observed here contradict the idea that cave-adaptations arise due to selection on genetic changes that occurs over longer periods. However, plasticity initially does not necessarily mean that these longer term processes are not at play, either on natural variation directly affecting the trait or genetic variation affecting plasticity.

B) The argument regarding canalization of pigmentation and melanophore number due to selection against plasticity seems flawed. I am not sure you can conclude anything about loss of plasticity regarding these traits, as they are not present to assess in the cavefish population in a way that was assessed in this study. If the authors would like to argue less plasticity in these traits, I would suggest using an alternative method to assess them (i.e. visualizing pigment cell numbers by L-DOPA or assessing eye size earlier in development when cavefish eyes are present).

C) The authors claim that due to plasticity it is possible to evolve adaptations in a short timeframe (Discussion, last paragraph). However, they looked in one generation at only 2 traits, and show modest effects on only one of these two. Thus, I would suggest that this is an overstatement.

3) Improved survival in darkness could be mediated by a number of factors, not just metabolic improvements. Are these survival effects stemming from one, or a complex cluster, of plastic phenotypes? Please address.

4) The authors claimed that surface morphs have "no obvious pre-adaptations to low light", but the reviewers are not sure they agree. Surface fish have been described in the literature as 'scotophilic', and many researchers have noted the propensity for surface fish, even in captivity, remaining in the darkest part of their tanks. The reviewers would further note that this information appears to be relevant to the interpretation of Figure 4 (elevated cortisol in constant darkness for SF). Please address.

---

## [Author Response]

Essential revisions:1) For a few traits, it appears that only one biological replicate was performed. These include cortisol levels, triglyceride levels, TSH levels and GH levels. It may be that I am misunderstanding the data. However, if this is the case, I recommend running more biological replicates. In line with this, the second generation survival results are rather mild. Can the authors provide information around the number of times this assay was performed, n values, etc.?

We answer these questions and provide clarification of these experiments as follows.

Cortisol and triglyceride levels were determined in 3 biological replicates. We originally had 2 biological replicates for the TSH and GH experiments. As a response to the reviewer’s critique we have made another biological replication of both determinations, and now have a total of 3 biological replicates. The new experiments replicated previous results that GH and TSH changed in SF under the influence of darkness and that levels of these hormones were different in CF and SF. The only difference in this experiment was that lower levels of TSH in D/D compared to L/D SF did not reach statistical significance (although they did show the same direction of change). For this reason we rewrote part of the Results.

The second generation survival experiment could only be performed once as it was very difficult to obtain embryos from fish raised to sexual maturity in the darkness, and the end result of the experiment is the death of larvae, so we could not repeat the experiment. Concerning the question about numbers, there were 24 larvae in each fish and treatment type at the beginning of the experiment. In total we used 96 larvae spawned in the darkness and the same number of fish spawned in L/D.

2) The reviewers are concerned that some of the conclusions are overstated in the Discussion. While the authors make a strong argument that many cave phenotypes can be phenocopied by phenotypic plasticity, they do not think from this study that claims can be made about natural selection acting to enhance or reduce plasticity. SpecificallyA) The initial sentence of the Discussion suggests that the plastic responses observed here contradict the idea that cave-adaptations arise due to selection on genetic changes that occurs over longer periods. However, plasticity initially does not necessarily mean that these longer term processes are not at play, either on natural variation directly affecting the trait or genetic variation affecting plasticity.

We agree with the reviewers and have changed the sentence in the Discussion as follows: “In addition to the widely accepted view that cave-adaptations result from long-term genetic processes (Barr, 1968; Culver, 1982; Juan et al., 2010), our results indicate that some cave-related traits can appear within a single generation by phenotypic plasticity.”

B) The argument regarding canalization of pigmentation and melanophore number due to selection against plasticity seems flawed. I am not sure you can conclude anything about loss of plasticity regarding these traits, as they are not present to assess in the cavefish population in a way that was assessed in this study. If the authors would like to argue less plasticity in these traits, I would suggest using an alternative method to assess them (i.e. visualizing pigment cell numbers by L-DOPA or assessing eye size earlier in development when cavefish eyes are present).

We agree with the reviewers and have removed all notions about canalization from the manuscript.

C) The authors claim that due to plasticity it is possible to evolve adaptations in a short timeframe (Discussion, last paragraph). However, they looked in one generation at only 2 traits, and show modest effects on only one of these two. Thus, I would suggest that this is an overstatement.

We agree and have removed that part of the sentence.

3) Improved survival in darkness could be mediated by a number of factors, not just metabolic improvements. Are these survival effects stemming from one, or a complex cluster, of plastic phenotypes? Please address.

We have addressed this issue and rewrote sentences accordingly in the Figure 3 legend and in the subsection “Starvation Resistance in Dark-raised Surface Fish”.

4) The authors claimed that surface morphs have "no obvious pre-adaptations to low light", but the reviewers are not sure they agree. Surface fish have been described in the literature as 'scotophilic', and many researchers have noted the propensity for surface fish, even in captivity, remaining in the darkest part of their tanks. The reviewers would further note that this information appears to be relevant to the interpretation of Figure 4 (elevated cortisol in constant darkness for SF). Please address.

We thank the reviewers for pointing out the issue of scotophila in *A. mexicanus* surface fish and have rewritten the sentence(s) claiming that *A. mexicanus* surface forms have no pre-adaptations to low light (Introduction). However, we intended to be more general in this introduction; and in the spirit of including other cave animals as examples have also pointed out that one cave dweller is known to be derived from a photophilic surface ancestor.

We agree that elevated cortisol in D/D SF suggests that the transition to a dark environment by surface ancestors of CF was a significant challenge with or without preadaptations, and this issue is addressed in the Discussion.